# Influence of Soil Chemical Features on Aromatic Profile of *V. vinifera* cv. Corvina Grapes and Wines: A Study-Case in Valpolicella Area (Italy) in a Calcareous and Non-Calcareous Soil

Diego Tomasi [1], Patrick Marcuzzo [1], Tiziana Nardi [1], Andrea Lonardi [2], Lorenzo Lovat [1], Riccardo Flamini [1] and Giovanni Mian [1,3,*]

1 Council for Agricultural Research and Economics—Research Centre for Viticulture and Enology, CREA-VE, Via XXVIII Aprile 26, 31015 Conegliano, Italy
2 Bertani Domains Società Agricola A.R.L., Via Asiago 1, 37023 Grezzana, Italy
3 Department of Agricultural, Food, Environmental and Animal Sciences, University of Udine, Via delle Scienze 206, 33100 Udine, Italy
* Correspondence: giovanni.mian@uniud.it; Tel.: +39-3478890182

**Abstract:** In the Valpolicella area (Verona, Italy) one of the most important Italian wines is being produced: Amarone, predominately made by *Vitis vinifera* cv. Corvina. This wine is subjected to a specific postharvest process, namely, withering, which aims to increase alcohol content and/or high residual sugars while retaining richness in aromas and organoleptic compounds. In accordance with guidelines and strict Amarone protocol set by the Consorzio of Amarone-Valpolicella, withering must be carried out by setting the grapes in a suitable environment. In our study, the withering process was performed following the traditional methods, i.e., in open-air, natural environmental conditions, as generally performed by winegrowers, leading to a weight loss in grapes of up to 30%. Whilst the effect of different postharvest conditions is clear and studied, it is interesting to know how soil can affect both grapes and wine profile, in terms of aromas, which are of great importance for this crafted wine. For this purpose, for two study years, the influence of two different sites (with regards to the carbonates' content) on aromas were investigated. Furthermore, microvinifications and the sensory profile of the resulting wines were analysed. Our results clearly indicated that different soils' characteristics, particularly differences in carbonate content, had an important effect, not only on the aromatic compounds of grapes and wines, but on the sensory profile as well. This information will help winegrowers and winemakers in the process of determining site selection for future vineyards in order to obtain a final optimum Amarone wine, in terms of its aromatic composition—one that is able to respond to the market demand.

**Keywords:** Amarone; polyphenols; soil features; sensorial analyses; VOCs

## 1. Introduction

Currently, a good vineyard is considered to be one that allows full ripening of selected grape varieties, achieving an adequate but not excessive vigour of the vines, showing optimum soil fertility and an adequate water supply, all of which together result in a complex, harmonious, and balanced final aromatic profile of the wine [1–4]. From this point of view, the production of high-quality grapes and wines is closely linked to soil, which is recognized as one of the most important aspects for the quality of grapes and wine [5]. Thus, the environmental factors, such as topographical, agro-pedological and climatic, along with soil features, are usually described by the French term "Terroir" [6]. This effect is an amalgamation of influences including climate, landscape (slope, exposure, and the biological and physical environment), soil, and geology [7]. In this scenario, many studies focused on the effects of climate [2,8], because it is considered to be the main constraining

issue. The influence of soil (texture, depth, chemical composition, fertility, and water availability) on the characteristics of a wine has also been studied in recent years [9,10].

In fact, soil provides the vines with nutrients and water, and any imbalance of them can affect vine growth and grape composition [11]. Due to the spatial variability of soil, even a short geographical distance can result in a significant impact. Furthermore, the texture of the soil has a great influence on vine and radical system development; consequently, the characteristics of the grapes and wines are affected [12]. Hence, soil geology (bedrock and overlying soils) can help to explain differences in grape and wine quality, even within the same area or climate classification [13–15]. In fact, the physicochemical properties of soil are determined by soil forming processes [16]. For example, carbonates' and active carbonates' content are properties that have an important influence on grapevine physiology and berry composition [17], to be seriously considered for Corvina grapes and Amarone wines [18]. Scarlet and co-authors [19], and Bramley and collaborators [20,21] confirmed that the spatial variability of rotundone concentration, responsible for the pepper aroma in Syrah and other red grape varieties, is associated with soil chemical properties and origin. It is also reported that in both Corvina and Corvinone cultivars, rotundone takes part in their aroma composition. Nevertheless, the precise dependence of the primary grape berry attributes in relation to the soil features (both physical and chemical) is still uncertain, as it is for the final wine profile, although it is reported that there is an important effect exerted by soil [5].

To date, winemakers are showing concern on the effects of soil composition and texture on grapes' quality and the final wine profile (especially with regards to the carbonates), especially in areas such as Valpolicella (Verona, Italy), because for market sales, a balanced presence of the organoleptic components of specific wines is required (Riesling, Vin Santo di Santorini, or Valpolicella Amarone). These latter are being cultivated in a limited geographical area, where the soil effect is stronger than elsewhere. In the Valpolicella municipality *Vitis vinifera* cv. Corvina is cultivated, and it is the main wine variety used to obtain, after grape withering, Amarone wine. In this procedure, grapes are generally withered until 30% of weight loss (WL) is reached [22]. Italy is one of the most significant worldwide producers of grapes and wines [23] and, within this sector, Amarone wine ranks in the top five in terms of both production and value. Moreover, Amarone is marked by organoleptic traits (aromas) such as cherry, bitter almond, and flower scents. For this purpose, it is important to determine the final level of aromas after the withering process, as well as the wine's sensory profile, of vines cultivated in different soils, due to the limited geographical are of cultivation.

This paper presents the results of a case study examining the existing association between environmental parameters (i.e., carbonates and related parameters in soil) and grapes' and wines' composition, regarding grapes that were withered postharvest, through the natural conditions in dehydrating cellars, until up to 30% WL. For this purpose, we conducted soil analysis and aroma extraction and quantification, together with microvinifications and sensory profile analyses. The vineyards presented identical climatic, topographic, and viticulture characteristics, and employed the same cultivation techniques, and the same withering and winemaking procedures, but the vines were planted in different soils. Given these circumstances, we presume there exists a significant connection between soil geology origin and differences between grapes and wines. The aim was to prove that wine can conserve the "gout de terroir", reflecting, in this way, its origin.

## 2. Materials and Methods

### 2.1. Experimental Setup

Commercial vineyards were selected in the classic Valpolicella area (loc. Novare in Negrar municipality, Verona, 45°32′44.9′′ N 10°57′26.6′′ E). The experimental data were collected during 2016 and 2017. The first vineyard (e.g., treatment) was called "Colombara" (C) and its geological origin is in sedimentary actions, whilst the second one was "Quaiara" (Q), geologically volcanic. The 2 vineyards were about 0.5 km apart; therefore, the climate conditions were exactly the same for the 2 study sites (data not shown—available at

avepa.it). The trellis system was Guyot, plant density was equal to $\pm 4500$ vines $ha^{-1}$, the variety was Corvina clones ISV-CV48 grafted onto Kober 5bb. Local standard practices were performed for pest management and fertilisation for both C and Q. Fertilisation was carried out in early spring and autumn, applying the same amount (220 kg $ha^{-1}$) of a balanced fertiliser (15N, 10P, 20K + Mg + S) in both study years. Plant-defence agrochemicals against pest and fungal disease were applied during the season, adopting commercial products. Plant agrochemicals were used in the same quantity for both C and Q vines, with the same number of treatments per year of study. The vines were irrigated according to the weather conditions to restore the field capacity. Experimental trial consisted in 3 random blocks (i.e., complete randomized block) for C and Q, in both study sites. The harvest period, as per local tradition, was set 10 days before commercial harvest for Valpolicella fresh wine. The fresh and selected grapes were stored in plastic boxes for natural conditions. Here, natural dehydration management was obtained with no air humidity control or mechanical air movement, but the supervisor had either to open or close the window, in order to help the air, rich in humidity, to be released, or to let the dry air enter. In detail, humidity (relative moisture in the air) was, on average, between 60 and 85% in function of outside conditions. Airflow was not quantifiable as the windows' opening was performed manually, only when the humidity was at too high of a level (e.g., more than 70%). Lastly, T° was, on average, between 14 °C and 20 °C at the beginning of experiments, going down to 6 °C to 12 °C at the end of experimentation. No record was performed concerning light intensity. The berry chemical composition was analysed at 30% WL, in accordance with the Amarone production law [22], since information (such as the entrance level in terms of aromas) for these grapes is entirely extraneous due to the fact that they are meant to be withered. Yet, after each withering process, vinifications were performed each study year; thus, the sensory profile was carried out.

### 2.2. Soil Site Characterisation

The soils of the experimental vineyards were characterised at the beginning of the experiment. Samples of approximately 1 kg of intact soil were randomly collected throughout the vineyard, each one consisting of 3 replicates. The samples were analysed by an external laboratory which applied standard methods, as indicated by Italian law (Law Decree n. 79/1992 and Law Decree n. 185/1999,) just adopted, and described by [24].

### 2.3. Analysis of Wine Must at the End of Withering, Polyphenols, and Glycosidic Aroma Compounds

Grape must parameters were measured with an automatic refractometer (Atago PR32) at 23 °C, following the literature [25], at the end of withering. Total berries' acidity expressed both as pH and titratable acidity (T.A.) (i.e., tartaric acid expressed as g $L^{-1}$ of tartaric acid) were measured using an automatic titrator (Crison Micro TT 2022) by titration with 0.1 N NaOH solution. The malic acid profile was determined by high-performance liquid chromatography (HPLC) (HPLC, ThermoFischer Scientific, Abingdon, UK). HPLC was equipped with a pump (PU980), a variable wavelength detector (UV970), set at 210 nm, and connected in series with a refractive index detector (RI830). Samples were injected with a 20 μL loop using a 7125 valve (Rheodyne Inc., Cotati, CA, USA). The separation was performed with an Aminex HPX-87H column (300 × 7.8 mm) protected with a pre-column (30 × 4.6 mm) filled with the same stationary phase (Bio-Rad Laboratories, Hercules, CA, USA). Column and pre-column were thermostatted at 45 °C by a heater (Jones Chromatography, Hengoed, Mid Glamorgan, UK). The conditions tested were as follows: flow, 0.5 mL/min; eluent, 0.003–0.05 N $H_2SO_4$ with 6% acetonitrile (*v/v*). Data acquisition and peak processing were performed with Borwin 5.0 software (JMBS Developments, Grenoble, France). Peak identification was based on retention times (Rt) and spiking technique, while peak quantification was based on the external standard method. The samples for HPLC were prepared by taking 250 μL of grape must diluted 1:50 with distilled water. The sample was then filtered through a 0.2 μm PTFE filter (Merck, Italy) and then HPLC analysed. The

grape must samples were prepared by pressing 3 subsamples (i.e., biological replicates) of 250 g of berries.

The aglycones liberated from glycoside-bound aroma precursors were analysed by gas chromatography/mass spectrometry (GC/MS, EI 70 eV, ThermoFischer Scientific, Loughborough, UK) after performing enzymatic hydrolysis. Full protocol adopted, previously described by us, is available in the literature. Shortly thereafter, total anthocyanin and aroma indexes were determined by following the method reported in the literature [22–26]. Analysis of glycosidic aroma compounds was performed by enzymatic hydrolysis and gas chromatography/mass spectrometry (GC/MS) of the aglycones liberated, according to the previous methods [22]. A 6850 GC/MS system was used (Agilent Technologies, Santa Clara, CA, USA), equipped with polyethylene glycol-fused silica HP-INNOWax capillary column (30 m × 0.25 mm, 0.25 μm), coupled with HP 5975C mass spectrometer and 7693A automatic liquid sampler injector. Identification of compounds was performed using NIST Mass Spectral Libraries Database (rev08) and the in-house CREA-VE database of grape and wine volatile compounds. Compound contents were expressed as μg of internal standard per kg dried grape (d.g.).

### 2.4. Winemaking and Tasting

The winemaking processes were performed via microvinification carried out at the vinification centre of the cooperative nurseries of Rauscedo (VCR), where 150–180 kg of dried grapes were used for each vinification (same process in each step), obtaining about 60 L of vine, as previously described by [27] and just adopted [22]. Wines were tasted 1 year following vinification, and were stored in stainless steel tanks in a temperature-conditioned cellar at 12–14 °C. Tasting analysis was performed in Conegliano (TV) at CREA-VE by a test panel made up of 10 properly trained specialist judges and a panel leader. To test and to confirm the reliability and accordance of judges, a periodic training among the participants to the panel test was carried out.

### 2.5. Statistical Analysis

One-way analysis of variance was performed using "R" free software (2021 version). Statistical analysis for the determination of significant differences between treatment means was carried out using Student–Newman–Keuls test ($p \leq 0.05$). Regarding the sensory profile, for the quantitative evaluation of the intensity of attributes (olfactory, gustatory-tactile, and retro-olfactory), quantitative descriptive analysis was used with the help of a question sheet providing discrete scale responses with intervals from 1 to 7. For each of the 3 attributes, the relative differences between wines were analysed and confirmed, submitting the judgements to statistical analysis using the ANOVA method (F-test $p$-values < 0.0001).

## 3. Results

### 3.1. Soil Classification of the Two Experimental Sites

Soil classification of the two experimental sites is reported in Table 1. Firstly, Quaiara is described as loam, whilst Colombara is described as clay loam, according to the United States' Department of Agriculture nomenclature. In fact, it is possible to note that significant differences were found in sand and clay content. Moreover, significant differences were found in the quantity of total carbonates and active carbonates; these were substantially higher in C than in Q. Specifically, total carbonates were almost 15-fold more in C in comparison with Q (45.02 and 3.03 $CaCO_3$, %, respectively). Active carbonates were almost sevenfold more in C than Q (13.90 and 2.10 $CaCO_3$, %, respectively). Considering the high carbonate content in C soil, it can be described as calcareous soil. Indeed, the ratio of Ca/Mg was significantly higher in C than Q (24.17 and 12.30, respectively). Finally, the ratio of Ca/K was again in favour of C (43.72 and 29.53, respectively).

**Table 1.** Soil characteristics of the two experimental sites (C (Colombara) and Q (Quaiara)). Within rows, *; **; *** values are different at $\alpha \leq 0.05$, 0.01, and 0.001. Data are the mean of three replicates.

| Experimental Site Characteristics | | |
|---|---|---|
| Parameter | Q | C |
| Sand (%) | 38.05 | 25.00 * |
| Silt (%) | 43.00 | 41.00 |
| Clay (%) | 19.04 | 34.00 * |
| Organic matter (g kg$^{-1}$) | 2.68 | 2.86 |
| Total nitrogen (N%) | 1.87 | 1.89 |
| pH (soil/water ratio= 1:2.5) | 7.01 | 7.86 |
| Total carbonates (CaCO$_3$, %) | 3.03 | 45.02 *** |
| Active carbonates (CaCO$_3$, %) | 2.10 | 13.90 *** |
| Available P$_2$O$_5$ (ppm) | 103.00 | 57.00 |
| Exchangeable K$_2$O (ppm) | 258.04 | 280 |
| Exchangeable MgO (ppm) | 285.00 | 231.00 |
| Exchangeable CaO (ppm) | 5782.08 | 5221.06 |
| Exchangeable Na (ppm) | 26.02 | 21.00 |
| Ratio Ca/Mg (meq) | 12.30 | 24.17 ** |
| Ratio Mg/K (meq) | 2.40 | 1.82 |
| Ratio Ca/K (meq) | 29.53 | 43.72 * |
| Ratio C/N | 8.31 | 8.78 |
| CSC (capacity of cationic exchange) (meq/100 gr) | 32.29 | 27.82 |
| Organic C | 1.55 | 1.66 |

The must parameters after the withering process are depicted in Table 2. No great differences arose between C and Q; hence, at the end of the dehydration, musts were fully comparable. Table 3 shows the anthocyanin content in C and Q, in 2016 and 2017. As can be noted, in 2016, the highest level was found in Q (886 mg Kg$^{-1}$). In 2017, again, the highest level was recorded in Q (441 and 335 mg Kg$^{-1}$, in Q and C, respectively).

**Table 2.** Wine must parameters after the withering process, in C (Colombara) and Q (Quaiara). Within rows, *; **; *** values are different at $\alpha \leq 0.05$, 0.01, and 0.001. Contents are expressed as mg Kg$^{-1}$.

| | 2016 | | 2017 | |
|---|---|---|---|---|
| | C | Q | C | Q |
| Sugar (Brix) | 32.1 | 33.2 | 25.4 | 27.5 |
| Acidity (g L$^{-1}$) | 7.7 | 7.2 | 8.7 | 8.8 |
| pH | 3.1 | 3.24 | 3.15 | 3.18 |
| Tartaric acid (g L$^{-1}$) | 6.4 | 6.1 | 9.2 | 8.9 |
| Malic acid (g L$^{-1}$) | 2 | 1.2* | 1.5 | 1.7 |

**Table 3.** Content of anthocyanins in 2016 and 2017, in C (Colombara) and Q (Quaiara). Within rows, *; **; *** values are different at $\alpha \leq 0.05$, 0.01, and 0.001. Contents are expressed as mg Kg$^{-1}$.

| Anthocyanin Content at 30% WL | | |
|---|---|---|
| 2016 | C | 614 |
| | Q | 886 ** |
| 2017 | C | 335 |
| | Q | 441 * |

### 3.2. Glycosidic Aroma Precursors

Glycoside aroma precursors identified in Corvina grape are reported in Table 4. Each class of compounds was calculated as µg IS/Kg d.g on the sum of normalized signal intensity of the single compounds belonging to the chemical class.

**Table 4.** Content of glycosidic aroma precursors in 2016 and 2017, in C (Colombara) and Q (Quaiara). Within rows, *; **; *** values are different at α ≤ 0.05, 0.01, and 0.001. Contents are expressed as µg 1-heptanol (IS)/Kg d.g. (dried grape).

| 2016 Harvest | 30% WL | |
| --- | --- | --- |
| | Q | C |
| Σ aliphatic alcohols | 491.85 | 621.19 * |
| Σ C-6-aldehydes | 34.84 | 35.05 |
| Σ monoterpenols | 721.41 | 997.10 * |
| Σ $C_{13}$-norisoprenoids | 1022.32 | 1326.45 * |
| Σ benzenoids | 4734.00 | 6340.39 ** |
| Σ furan derivates | 137.72 | 105.87 |
| TOTAL | 7142.14 | 9416.05 ** |
| **Harvest 2017** | **30% WL** | |
| | Q | C |
| Σ aliphatic alcohols | 760.35 | 850.05 * |
| Σ C-6-aldehydes | 73.48 | 120.23 ** |
| Σ monoterpenols | 977.13 | 854.16 |
| Σ $C_{13}$-norisoprenoids | 1458.69 | 3324.27 *** |
| Σ benzenoids | 6133.46 | 9979.90 ** |
| Σ furan derivates | 68.92 | 85.05 * |
| TOTAL | 9472.02 | 15,213.67 ** |

In both years tested, the total content of aroma compound precursors was almost always higher in C than Q. In 2016, total aliphatic alcohols, total monoterpenols, and total $C_{13}$-norisoprenoids and benzenoids were the highest in C. Aliphatic alcohols account for herbaceous and unripe fruit aroma, and the wine tasting can reveal their presence. Monoterpenols confer floral or citrus notes to the wines, whilst $C_{13}$-norisoprenoids are correlated to floral/spicy notes arising in red wines after ageing, and potentially constituting precursors of volatile compounds which, in turn, contribute by conferring positive notes to wines' aroma (red ripe fruit). Regarding benzenoids, once again, the content was higher in C than Q; this can contribute to the typical aroma of Amarone wines. Concerning furans, no significant differences were found between C and Q. In 2017, the content of aliphatic alcohols, benzenoids, C-6-aldehydes (which confer herbaceous/grassy notes to the wines), $C_{13}$-norisoprenoids, and furan derivates were higher in C than Q.

### 3.3. Wine Tasting

After two study years, the winemaking was performed. The result of wine tasting conducted by a panel of experts is listed in Figure 1. A statistical difference can be observed for the smooth profile of wine, highlighting a higher smoothness for the wine from Colombara than that from Quaiara, along with a lesser herbaceous hint and a better taste balance ($p < 0.05$). Concerning the other parameters, although no statistical differences were reported on average, Colombara wines presented an increased taste of red ripe fruit, with a better taste balance and richness complexity. As for 2017, the wine judgment panel confirmed the better aroma performance of C wine compared to Q, while Q scored higher points in terms of body, probably due to the higher sugar content of the dried grapes. These parameters seem to highlight an effect of soil profiles on the final products, allowing one to obtain the richest and more complex wine when viticulture is performed in calcareous soils (Colombara case). This is also confirmed by global wine appreciation, where Colombara has received a significantly higher score than Quaiara ($p < 0.05$).

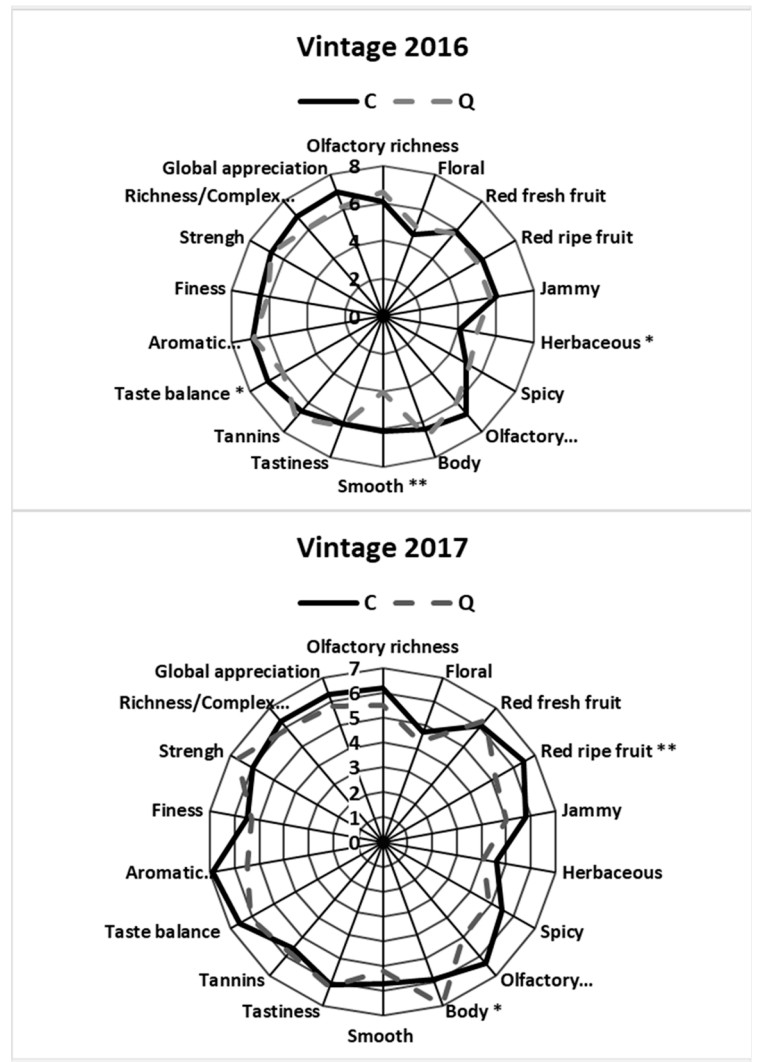

**Figure 1.** Sensory profiles of wines in Colombara (C) and Quaiara (Q) in the years 2016 and 2017. Where present, *; **; *** indicate a significant difference at α ≤ 0.05, 0.01, and 0.001.

## 4. Discussion

Amarone wine is mainly obtained from the Corvina variety; grapes are harvested and stored in dehydrating rooms before the vinification is performed. This is a specific process, adopted to obtain reinforced wines, hence, richest in alcohol, anthocyanins, flavour, and structure [28–30]. In this context, and for all wines, the starting point of grape composition is of particular importance in order to obtain a final good quality of grapes and wine profile— one that is richer in aromas. The accumulation of substances in grapes is determined by various aspects. In fact, grape and wine quality are often associated with specific conditions of soil, climate, cultural practices, and training systems, leading to the significance of the terroir. In this sense, soil features lead to different wine profiles [31,32] and are recognized as the most important environmental factor affecting vine development and wine quality. Carbonates play an important role. In this regard, various studies were carried out on the involvement of soil features on grape and wine composition [9,31,33,34].

In our research, two vineyards planted on distinct soil types (i.e., carbonate content) were used to study the soil's effect on the composition of Corvina grapes and Amarone wine, in terms of aromas. Firstly, we can report how the two soils were different, mainly regarding the carbonate amount. It was decided to test these two soil conditions—rich in carbonates and not rich—due to their importance in plant nutrient uptake [7,18,35,36]. Thus, among all the features of soil found by analyses, we report that the greatest difference

was determined by carbonate quantity and related indices. This is a key factor for our study: as reported in the introduction, the soil's physical properties and carbonate chemical character were correlated and affected the nutritional status of the vine leaves, resulting in completely different physiological responses by the plants.

Beginning with the aforementioned conclusion, we investigated, in two study years, grape composition in terms of anthocyanins content, aroma compound, and wine sensory profile. In the final analysis, the must parameters, such as sugars, pH, acidity, etc.—with the except of malic acid—were not influenced by soil. After the withering process, they accounted for the same level. Only malic acid was affected, as has been reported in the literature [7]. Poor soils, along with soil geochemistry, usually promote smaller yields of grapes, with more concentrated flavours, while fertile soils lead to overgrown vines, which have a significant impact on berry quality (optimum berry quality is seldom achieved if vines are excessively vigorous).

Plant phenolics have important effects on food quality and human nutrition. Their presence in grapes and red wines contributes to health benefits because of their antioxidant and anticarcinogenic activities. Furthermore, the biosynthesis of antioxidant molecules such as anthocyanins represents a defence mechanism for the plant cells. These molecules have a beneficial impact on humans who consume them through drinking wine. Anthocyanins are the second class of phenolics, of utmost sensory and oenological importance, after tannins, and, in our experience, in both study years, they were higher in Q than C (Table 2). Hence, further investigation may be needed for this analysis; however, as explained above, the differences in 2016 could be determined by the growing season itself. In any case, our explanation for higher anthocyanin concentrations in grape skins in Q could be the determined by differing vine status, lighter berry weight, and heavier skin weight at harvest (data not shown). This is surely related to soil condition; thus, grapevine cv. Corvina cultivated in Q conditions generally accumulates more anthocyanins, which does not promote yield and yield-related parameters, as reported in the literature [27].

We took into consideration and analysed the most important aromas classes involved in the aromatic profile of grapes and final wines (Table 3). According to the literature [21,37], the scent of Amarone recalls dried fruit, cherry, and spices. Norisoprenoids are the most important family of aromatic compounds which provide wine noticeable notes of red fruit (es β-damascenone), combining with furan derivates. In 2016, the total aroma content was higher in C than Q. Specifically, the aliphatic alcohols, monoterpenols, and $C_{13}$-norisoprenoids were the single class with the highest values in C. The same trend was confirmed by the investigation carried out in 2017, where aliphatic alcohols, C-6-aldehydes, $C_{13}$-norisoprenoids benzenoids, and furans were the class accounting for the highest levels in C. More importantly, the most representative aroma classes' amount was in accordance with the final wine profile. It is possible to conclude that soil carbonates greatly influenced aromas in C grapes, since they could influence wine character due to soil–grapevine interactions. In fact, Carbonates, pH, and Ca content are important soil features, acting as the coordinating ion of chlorophyll, as well as activating many enzymes needed for plant vitality. In this sense, our study improves our understanding of how geological variations impact plant biological processes and, in turn, account for flavour dichotomy.

Furthermore, in accordance with the aforementioned instances, concerning the sensory evaluations (Figure 1), the wines that we obtained confirmed the differences found in all analyses conducted on grape aromas. C wines achieved a greater smoothness, olfactory pleasantness, persistence, and generally red ripe fruit scents. The panel also found C wines more balanced and more harmonious overall. On the contrary, due to the higher grape sugar content in Q, the wine appeared more lightly structured with a more evident fullness. However, increased structure of Q is also due to more anthocyanins and, of course, not only due to more sugars. We can conclude that carbonates affected this parameter. It is well known that carbonate levels increase the amount of chlorophyll, which impacts primary metabolism (acids and sugar) and plant growth, produces higher crop yield, and improves

the optimal grape ripeness [38]. This latter, superior grape ripeness is our final explanation for the greater aroma and wine sensory profile in C.

Given these data, soil features, and—particularly—carbonate quantity strongly influenced aroma accumulation. This is in agreement with what was found previously in other types of studies [39]. In previous studies, authors explained how high carbonate soil level reduces dry matter; hence, less production may lead to a major accumulation of aromatic compounds in berries. In our study, the same occurred: a reduction in total yield in C was recorded (data not shown). This could have led to a better quality of grapes and related musts/wines in C, since plants promoted the activity of many biochemical pathways instead of having to produce green matter (e.g., yield). Additionally, it should be noted that limestone generally provides good drainage [40], helping to avoid anoxia in roots, improving vine physiology, and generally leading to better vine health.

This was confirmed by tasting the final wines. The global appreciation was significantly higher in the wine obtained from the calcareous area and, again, the better organoleptic profiles were achieved in the calcareous soil, in accordance with the aroma classes found. The literature, in this sense, has reported the propensity to obtain higher aromatic wine when higher limestone content is present [41,42].

## 5. Conclusions

In general, the calcareous soil led to a higher content of aromas in C compared to Q, confirmed by the more interesting aromatic and elegant wine profile of these wines. Finally, considering that the impact of the lime footprint is quite clear, leading to a considerable content of bio-compounds and resulting in balanced wines, we can report that it is desirable to utilize soils such as those found in C. The sensory evaluation of wines highlighted the positive sensory descriptors present in the wines produced in C soil and the stronger character of the wines produced in Q. As a final consideration, we would like to underline how—as is generally acknowledged—soil characteristics, alongside agronomic practices and the environment, are of absolute importance for high-quality wine production. In order for winegrowers and winemakers to achieve their goals, soil and carbonate features should be seriously considered before planting vineyards, particularly for distinctive wines such as Amarone.

**Author Contributions:** Conceptualization, D.T. and G.M.; methodology, L.L., R.F. and T.N.; software, G.M. and R.F.; validation, G.M., D.T., R.F., T.N. and P.M., formal analysis, T.N., L.L. and R.F.; investigation, D.T.; resources, D.T., G.M., P.M. and A.L.; data curation, D.T., G.M. and R.F.; writing—original draft preparation, G.M. and P.M.; writing—review and editing, G.M., D.T. and R.F.; visualization, G.M.; supervision, D.T.; project administration, D.T.; funding acquisition, D.T., G.M. and A.L. All authors have read and agreed to the published version of the manuscript.

**Funding:** This research was privately funded by "Bertani Winery".

**Data Availability Statement:** Not applicable.

**Acknowledgments:** Authors thanks Bertani Domains for funding the research, and for allowing us to conduct the experiments in their vineyards and withering rooms giving all the necessary support.

**Conflicts of Interest:** The authors declare no conflict of interest.

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
