# Peer review of "Influence of Soil Chemical Features on Aromatic Profile of V. vinifera cv. Corvina Grapes and Wines: A Study-Case in Valpolicella Area (Italy) in a Calcareous and Non-Calcareous Soil"

_agriculture, doi:10.3390/agriculture12121980_

Round 1

Reviewer 1 Report

Comments
The manuscript "Influence of soil chemical features on the aromatic profile of V. vinifera cv. Corvina grapes and wines: a case of study in Valpolicella area (Italy) in a calcareous and non-calcareous soils " represents an attractive work with reproductive statistical analysis methods. However, the authors must make some modifications before considering the paper for publication.

1.      Lines 98-99

Is 0.5 Km enough distance to consider the two vineyards are different in terms of soil?

2.      Lines 116-118

Even though the dehydration is done naturally, the authors must provide the dehydration conditions like temperature, light level, airflow, and humidity.

3.      Line 121

The authors start the vinification process after two years of harvesting the samples! That could demolish the whole theory of the manuscript because that greatly affects the aroma /volatile compounds and other parameters. The authors should make a justification for that.

4.      Line 135-136

The definition High- Pressure Liquid Chromatography must change to High- Performance Liquid Chromatography.

5.      Lines 136- 137

The authors must provide the HPLC conditions (Flow rate, Column type, Temperature, and elution system).

6.      Line 259

The discussion section needs more explanation.

Author Response

Dear reviewers. We would like to thank you for the kind work performed so as to improve the manuscript. We found every comment really useful. In this text we are answering you back yet reporting into the main text the changes in accordance with you all (yellow highlighted).

On the behalf of all authors, thanks a lot again.

The corresponding, Giovanni Mian

Reviewer n.1:

  1. Lines 98-99

Is 0.5 Km enough distance to consider the two vineyards are different in terms of soil? Dear reviewer. We can confirm it by soil analyses of course. Not only, dated back to 1920, there is an annual review where authors reported the soil origin of the 2 sites. Unlucky, where are not able to recover it anymore.

  1. Lines 116-118

Even though the dehydration is done naturally, the authors must provide the dehydration conditions like temperature, light level, airflow, and humidity. Dear review, you are right. We added this information as much detailed as we could.

  1. Line 121

The authors start the vinification process after two years of harvesting the samples! That could demolish the whole theory of the manuscript because that greatly affects the aroma /volatile compounds and other parameters. The authors should make a justification for that. Dear review, you are right. There is a huge mistake made by us during the writing. Of course, as it is possible to see by the results, vinifications were performed in each year of study at the end of withering process. Changed in the main text.

  1. Line 135-136

The definition High- Pressure Liquid Chromatography must change to High- Performance Liquid Chromatography. Agree, changed.

  1. Lines 136- 137

The authors must provide the HPLC conditions (Flow rate, Column type, Temperature, and elution system). Agree, we added, thanks

  1. Line 259

The discussion section needs more explanation. Thank, we tried to improve it as much as we could. Hopefully now it is much better.

Authors would like to thank the kind anonymous reviewer once again for the great effort.

Reviewer 2 Report

Dear authors please check the details below

Line 41: “such are” change it to such as.

Line 42: “the soil ….” Remove “the”.

Line 46: “considered’” add “to be” after considered.

Line 89: I suggest you to change “assumed” to other formal words

Line 99: “distant” add distant apart.

Table 1: at least in the table’s note explain the Q and C abbreviations.

Line 315-322: This section needs to explain more. I would like to know why and how the soil affect the wine aroma.

My main concern is that; the hypotheses of the research wasn’t answered properly. I was looking for more details about the impact of soil characteristic on wine aroma. The discussion part should have more explanation. 

Thanks for your great work!!

Author Response

Reviewer n.2:

Dear authors please check the details below

Line 41: “such are” change it to such as. Thank you dear reviewer, changed.

Line 42: “the soil ….” Remove “the”. Thanks, done

Line 46: “considered’” add “to be” after considered. Thanks, done

Line 89: I suggest you to change “assumed” to other formal words. We agree, great thanks

Line 99: “distant” add distant apart. We agree, great thanks

Table 1: at least in the table’s note explain the Q and C abbreviations. Yep, we totally agree. Sorry for the forgetfulness

Line 315-322: This section needs to explain more. I would like to know why and how the soil affect the wine aroma. Yes, we agree, thank you

My main concern is that; the hypotheses of the research wasn’t answered properly. I was looking for more details about the impact of soil characteristic on wine aroma. The discussion part should have more explanation. Dear editor, we agree with improving the discussion section, also the other reviewer arose this point. We did at our max capability. Great thanks

Thanks for your great work!!

Thanks again to the kind reviewer for the great job made. We really appreciated it

Round 2

Reviewer 2 Report

Dear authors,

I appreciate your effort and time making the changes. The article looks much butter now.

Author Response

On the behalf of all authors, thanks a lot again.